# A Sensing Device with the Optical Temperature Sensors-Based Quad-RX Module and a Security Module

**DOI:** 10.3390/s21051620

**Published:** 2021-02-25

**Authors:** Kyeeun Kim, Siwoong Park, Chanil Yeo, Hyoung-Jun Park, Hyunjin Kim, Young Soon Heo, Hyun Seo Kang, Kyungsoo Kim

**Affiliations:** 1Optical ICT Convergence Section, Electronics and Telecommunications Research Institute, Gwangju 34129, Korea; kyeeun_kim@etri.re.kr (K.K.); swp@etri.re.kr (S.P.); ciyeo@etri.re.kr (C.Y.); spacegon@etri.re.kr (H.-J.P.); hjkim80@etri.re.kr (H.K.); ysheo@etri.re.kr (Y.S.H.); 2IMR, Gwangju 61008, Korea; imr@imrbiz.co.kr

**Keywords:** sensor node, optical receiver, real-time monitoring, fiber Bragg grating, security module, symmetric-key algorithm, secure hash algorithm

## Abstract

In this paper, we present a sensing device with the optical temperature sensors-based quad receiver (Quad-RX) module and a security module. In addition, in order to prevent cyberattacks on critical national infrastructures and key facilities, we implemented symmetric-key and secure hash algorithm-based hardware security modules in the key elements of the sensing device. A preliminary test was conducted prior to a field trial to verify the performance of the developed sensing device. The accuracy and stability of the sensing device were then verified for 1 month in a field test at facilities for energy storage systems and photovoltaic converters in sewage treatment plants.

## 1. Introduction

Over the last decade, real-time facility monitoring has attracted significant attention as a promising method for identifying various hidden risks and reducing energy loss in important facilities, such as power plants, electric power substations, energy storage systems (ESS), and civil structures [1,2,3,4]. In real-time facility monitoring, sensor nodes equipped with various electrical and optical sensors for detecting temperature, illuminance, humidity, strain, and vibration are typically used to collect information regarding the target facility. In recent years, cyber security has become increasingly crucial for preventing cyberattacks on critical national infrastructures and key facilities. Therefore, sensor nodes integrating security modules are essential for such cases [5]. Security modules are also necessary in the gateway and control platform, constituting a facility monitoring system to ensure comprehensive protection.

In this paper, we present a sensing device with the optical temperature sensors-based quad receiver (Quad-RX) module. In particular, the optical temperature sensor employs a fiber Bragg grating (FBG) sensor to collect a large number of temperature data at various points simultaneously, using multichannel grating [6,7]. In addition, the FBG sensor satisfied the required temperature accuracy of ±1 °C. The FBG sensor can detect temperature changes by linearly shifting peak wavelength by 0.01 nm when a 1 °C change occurs [8]. Therefore, the FBG sensor, which is a passive sensor and thus does not require a power supply, is not affected by electromagnetic interference (EMI) and thus can be used in electrical facilities [9,10]. Specifically, we utilized a quad-receiver module that was developed to reduce the volume and production cost of the FBG-based sensor [11]. To protect the fragile sensing apparatus that is attached, we packaged the optical temperature sensor in a brass tube.

This sensing device also contains a symmetric-key algorithm and secure hash algorithm (SHA)-based security module, which has advantages in terms of its simple structure, easy implementation, and high processing speed [12]. Hardware-based security modules have various advantages over software-based security modules, such as secure storage and management of encryption keys and rapid encryption/decryption processing. Therefore, we programmed the security system to be highly secure and to selectively use four methods (i.e., the academy, research, institute, agency (ARIA), high security and light weight (HIGHT), SEED (128-bit symmetric block chipher), and SHA) that can be changed in seconds for each cycle (as opposed to a single method). The encrypted sensor data was thus transmitted, meeting the required security requirements.

We conducted temperature sensing with the developed sensing device with the optical temperature sensor-based Quad-RX module. The calibration temperature was very close to the sensing temperature, with a difference of only ±1.6 °C. Thereafter, reproducibility was confirmed through repeated temperature experiments. On average, the Quad-RX module showed a reproducibility of 0.4 °C and the FBG sensor had an average peak wavelength shift of 0.01 nm. Preliminary testing was performed prior to field trials to verify the performance of the sensing device and gateway. A field test was then conducted at ESS facilities and photovoltaic converters in sewage treatment plants for 1 month.

## 2. Experimental Details

### 2.1. Design of the Proposed Optical Sensing Device

Figure 1 presents a schematic diagram of the sensing device with a double-layered structure. The first layer consists of a signal-processing interface circuit for connecting the optical temperature sensors, a broadband light source (BLS), and Quad-RX. In this study, a BLS with emission wavelengths ranging from 1530 to 1550 nm was utilized as a transmitter for the FBG-based temperature sensor. The BLS provides several benefits, including high stability and the capability of increasing the point of sensing [13]. In this paper, the point of sensing means the number of gratings, that is the number of measurable sensors. Another sensing device obtained the data with a tunable laser diode (LD) capable of selecting a wavelength [14]. In this study, in order to use a relatively inexpensive BLS, the bandpass filter and the linear transmission filter were used to select a wavelength and sense the corresponding wavelength region data.

The second layer consists of an interface circuit for connecting electrical sensors, a microcontroller unit (MCU), a security module, and two wireless communication modules (Wi-Fi and Bluetooth). These wireless communication modules were used to transfer the data between sensing device, gateway, and control platform. To present the sensor data and communication state, a liquid crystal display (LCD) and illumination sensor were installed on the top cover. An electrical sensor interface circuit includes electronic sensors (i.e., humidity, angular velocity, and acceleration) and makes these electronic sensors and the illumination sensor on the top cover of the sensing device operate normally and transmit data. These electrical sensors can be usefully used depending on the measurement environment. For example, when a sensing device is installed on a power pole, tilt or vibration is measured by an acceleration or angular velocity sensor to prevent safety accidents.

Figure 2a presents a schematic view of a Quad-RX module, which was used to collect optical signals from the FBG sensor. This module consists of a main body, four p-i-n (PIN) photo diodes (PDs), three beam splitters (BSs), two linear filters (LFs), two band-pass filters (BFs), and an optical fiber. The BSs split a received beam into a 5:5 ratio and divide the optical path in the Quad-RX module. Figure 2b is a graph showing the transmittance of the BF and LF as the current received from the PD. The LFs and BFs selectively pass wavelengths ranging from 1535 to 1539 nm into PD_1_ and PD_2_ and from 1545 to 1549 nm into PD_3_ and PD_4_. The measured photocurrents of PD_1_, PD_2_, PD_3_, and PD_4_ (i.e., R_1_, R_2_, R_3_, R_4_) were used to convert received optical signals into electrical signals. By dividing the R_1_ (R_3_) output by the R_2_ (R_4_) output, the peak wavelength shift can be calculated [15]. The peak wavelength shift presents the change in external physical quantities, i.e., temperature change in this paper [6]. It is worth noting that this calculation can also extract strain information using the FBG sensor [16]. Because the Quad-RX module that senses the data from the temperature sensor is contained in the sensing device, and the device is usually used at room temperature independently of the temperature sensor, the temperature stabilization of the LF or BF is not considered herein.

Figure 3a,b present the developed symmetric-key algorithm and SHA-based security module for our real-time facility monitoring system and a test setup for evaluating the security module. The security module contains an MCU (M4 core at 84 MHz, ARM^®^ Cortex^®^) and was designed to provide symmetric-key-based encryption and decryption of up to 64 bytes (512 bits) of plain text using the academy, research, institute, agency (ARIA), high security and light weight (HIGHT), and SEED algorithms [17,18]. Additionally, the security module was designed to provide a cryptographic hash function (i.e., SHA), which condenses input data into fixed size outputs [19]. It is worth emphasizing that this security module was developed to increase the number of encryption and decryption methods (e.g., public key algorithms and digital signature algorithms) to enhance the security of the real-time monitoring system. Table 1 lists the possible encryption types and key information for the developed security module.

To pass encrypted data from a sensing device to a control platform, we developed a gateway with a security module. This gateway was designed to minimize electromagnetic interference (EMI) and to connect up to 12 sensing devices to reduce the overall volume of the monitoring system [9,10]. For the further extension of sensing devices, we designed the gateway to accept wireless communication modules in a pluggable form. The entire procedure for data transmission from the sensing device to the control platform requires at least 3 s (including encryption and decryption). Therefore, we programed the security system to alternatingly apply the ARIA, HIGHT, SEED, and SHA algorithms every 3 s and transmit the encrypted sensor data using wireless communications (i.e., Wi-Fi and Bluetooth).

### 2.2. Fabrication

Figure 4 presents the produced Quad-RX module. This Quad-RX module integrates four analog PIN PDs in the main body and employs a ferrule connector (FC)/angled physical contact (APC) connector. Additionally, filter holders and thin-film filters, including BSs, LFs, and BFs, were integrated in the main body, as shown in Figure 2. The main body was produced using a laser welding method. Precise optical alignment of the four PIN PDs (PD_1_, PD_2_, PD_3_, and PD_4_) was conducted using an active alignment process after connecting the input fiber to the main body.

Figure 5 shows the optical temperature sensor using the FBG sensor made of bare fiber. When an optical sensor in the form of bare fiber is directly attached to the facility, it may break. Therefore, we packaged the sensing component in a brass tube with a good thermal conductivity. The sensing component was packaged in a brass tube and fixed. At the time, if the brass tube was too short or hard, epoxy was used and the optical temperature sensor was pulled tight. Tight packaging breaks the period of a grating designed with a certain period to reflect a specific wavelength. This led to a low linearity because it is hard to reflect the wavelength correctly when conducting temperature sensing. After packaging, the optical temperature sensor employed a ferrule connector (FC)/angled physical contact (APC) connector as the input fiber.

Figure 6a,b present top-view images of the produced doubled-layered sensor board. As mentioned previously, the first layer integrates a BLS, a Quad-RX module, a signal processing component, and an optical interface component. The second layer contains an MCU, electronic sensors, a security module, an electronic sensor interface, and Wi-Fi and Bluetooth modules, as shown in Figure 6b. Figure 6c presents the end product covered with a plastic case with a LCD panel.

### 2.3. Experimental Results

Figure 7 shows the output characteristics of packaged optical sensors 1 and 2 measuring the peak wavelength by increasing the temperature from 0 °C to 100 °C in a chamber with constant temperature and humidity (SU-641, ESPEC), measured using an optical spectrum analyzer (AQ6319, YOKOGAWA). The optical temperature sensor R-square (Coefficient of Determination, COD), which is a linear factor, was 0.99. We confirmed that the peak wavelength, detected by the optical temperature sensor, shifted linearly 0.01 nm according to a 1 °C change. Since the FBG sensor can only receive wavelength information, a calibration process was required to convert the wavelength to temperature [10]. We conducted a temperature calibration process for the produced FBG-based temperature sensor by placing the sensing device in a chamber with a constant temperature and humidity. In this study, we only utilized the measured photocurrents of PD_1_ and PD_2_ (R_1_ and R_2_) to calibrate the FBG-based temperature sensor.

Figure 8a presents the measured photocurrents and R_1_/R_2_ values for the FBG-based temperature sensor as a function of the temperature in the chamber. When raising the temperature, R_1_ and R_2_ fluctuated. However, since the fluctuation trend was similar, the R_1_/R_2_ value did not affect the temperature detection in terms of linearity [11]. We calculated R_1_/R_2_ for the temperature calibration and the results are plotted in Figure 8a. After plotting the R_1_/R_2_ over temperature, polynomial fitting was performed for calibration. From these results, we obtained R_1_/R_2_-dependent temperature data and derived the calibration temperature from R_1_/R_2_, as shown in Figure 8b. The calibration temperature was very close to the actual temperature, with a difference of only ±1.6 °C. Temperature experiments were repeatedly performed to confirm the reproducibility, with the Quad-RX module demonstrating an average reproducibility of 0.4 °C and the FBG sensor demonstrating an average peak wavelength shift of 0.01 nm.

Figure 9 presents the processes for the encryption and decryption between the sensing device and gateway. In all of these processes, Wi-fi or Bluetooth was used for packet transmission. In Figure 9i, the packet shows that the user had to enter into the security module for encryption. The blue box shows the algorithm type, i.e., the ARIA algorithm. The green box presents the data for encryption, i.e., “61 62 63 64 … 8f 00”, and the encrypted data are shown in the red box. In addition, if the encrypted data are transmitted to the security module of the gateway, as shown in Figure 9ii, a decryption packet (green box) and a decrypted packet (red box), as shown in Figure 9iii, are created. Here, “Check sum” is the process of checking the integrity of the encryption data using the SHA algorithm. When the decrypted data, i.e., “61626364…8f00”, are the same as “the data for encryption”, the user can finally check the monitoring data.

Figure 10 presents encryption (encoding) data transfer and decryption (decoding) between the sensing device and the gateway. Figure 10a shows the encryption and transmission of the original value on the sensing device to the gateway, and Figure 10b shows the decryption of encrypted data on the gateway. In the sensing device, the original values of temperature and humidity are encoded by ARIA and send to gateway. The gateway also sends the data to the control platform in the same way and the user can check the monitoring data. On the basis of this process, we can confirm that encryption and decryption using the developed security module work reasonably well. In this security procedure, data other than the communication pin are not exposed, so it is impossible for an attacker to steal the packet from the security module. In addition, even if it is hacked, it cannot be decrypted because the attacker cannot obtain the encryption key.

Figure 11a presents the proposed security-enhanced control platform, which was developed using a constrained application protocol (CoAP)-based lightweight machine-to-machine (LWM2M) protocol for real-time facility monitoring. For the field test, we installed sensing devices, a gateway, and optical temperature sensors at ESS facilities and a photovoltaic (PV) converter in a sewage treatment plant for 1 month, as shown in Figure 11b,c. Two optical temperature sensors are attached to the facility, and these sensors are connected to the sensing device through a FC/APC connector. These two sensors monitor the temperature at two points through the two wavelengths reflected from each grating. In this paper, two channel gratings, including two gratings in one fiber, are used. But if more gratings are used, the temperature at more points can be measured.

As shown in Figure 11a, the temperature, humidity, illuminance, angular velocity, and acceleration data collected from the sensors inside the sensing devices could be visually monitored using the control platform in real time. On the basis of this field test, we can confirm the accuracy and stability of the developed security-enhanced sensing device.

## 3. Conclusions

In this paper, we present a sensing device with an optical temperature sensor-based Quad-RX module and a symmetric-key algorithm and SHA-based security module. The sensing device was designed to measure temperature using optical temperature sensors. The optical temperature sensors attached to the facility and a Quad-RX module contained in the sensing device were used measuring the temperatures at two points by reflected wavelengths. If the number of gratings of the optical temperature sensor is increased according to the usage environment, it is possible to measure the temperature in more points. In addition, the optical sensor, which is a passive sensor and thus does not require a power supply, is not affected by EMI and its wavelength reproducibility is on average 0.01 nm. The temperature reproducibility of the Quad-RX module is on average 0.4 °C, and the calibration temperature was very close to the actual temperature with a difference of only ±1.6 °C. This sensing device can also measure illumination, humidity, angular velocity, and acceleration through electronic sensors in the device. Additionally, the security module was developed to be easily extendable for further improvements to the security of the sensing device. The security system was designed to use the ARIA, HIGHT, SEED, and SHA algorithms for the encryption of sensor data every 3 s and to transmit the encrypted sensor data. On the basis of the preliminary tests and the field trial, we confirmed the accuracy and stability of the proposed security-enhanced sensing device. Therefore, the proposed sensing device can be applied in critical national infrastructures and key facilities.

## Figures and Tables

**Figure 1 sensors-21-01620-f001:**
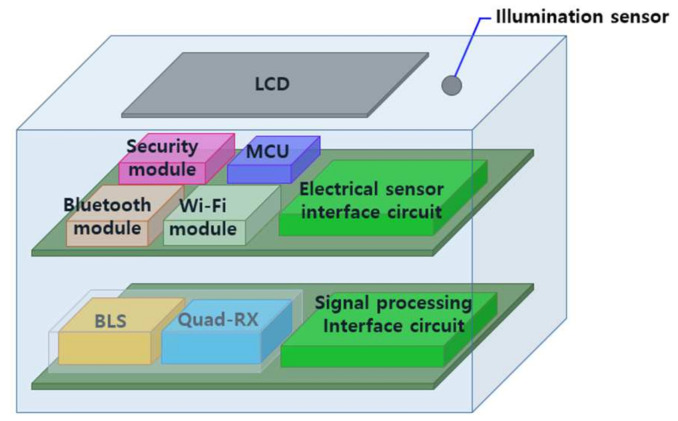
Schematic diagram of the sensing device.

**Figure 2 sensors-21-01620-f002:**
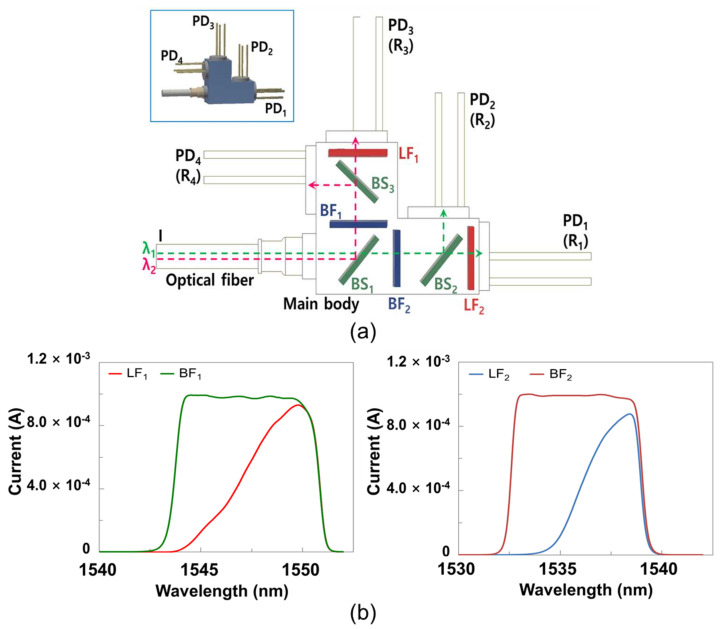
(**a**) A schematic view of a quad receiver (Quad-RX) module and (**b**) the current output characteristics of the linear filter (LF) and band-pass filter (BF).

**Figure 3 sensors-21-01620-f003:**
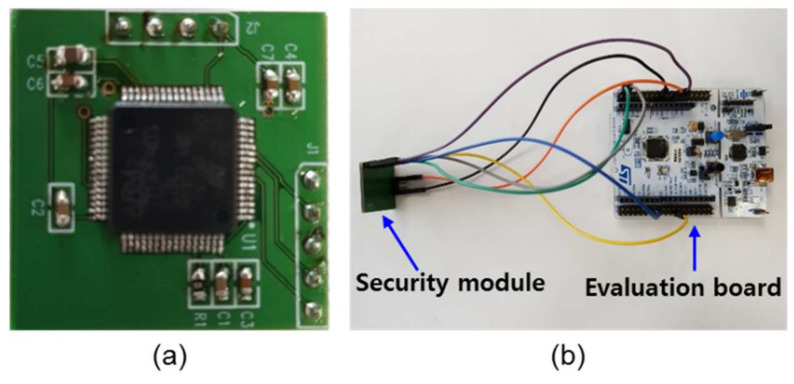
Photographs of (**a**) the developed security module and (**b**) the evaluation setup.

**Figure 4 sensors-21-01620-f004:**
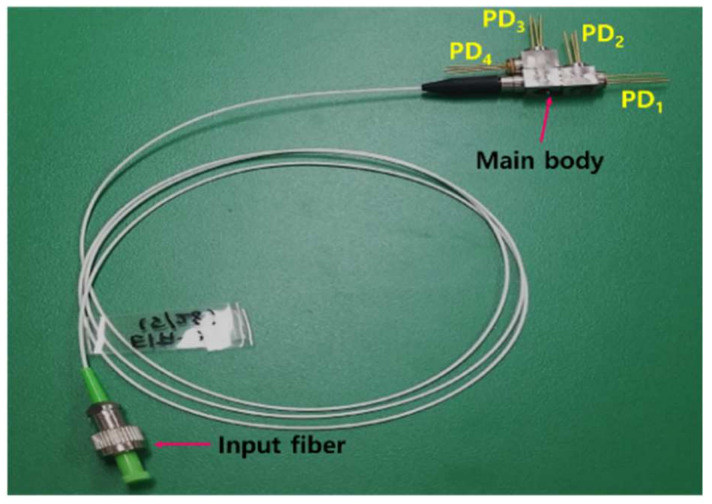
Photograph of a packaged Quad-RX module.

**Figure 5 sensors-21-01620-f005:**
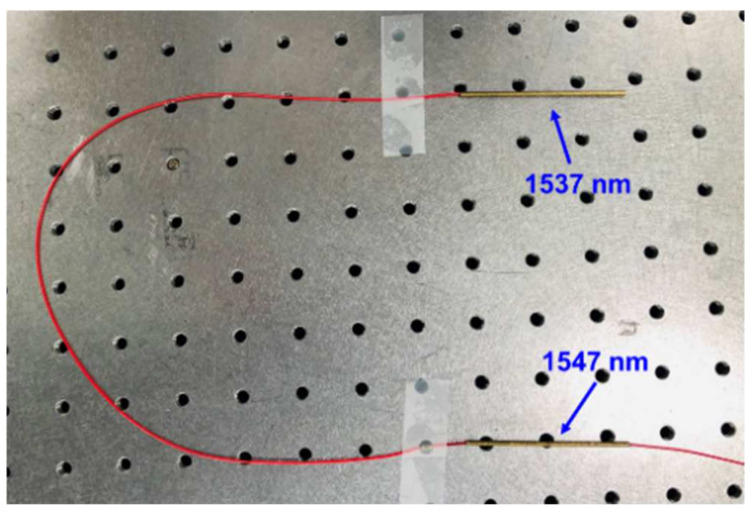
Photograph of a packaged optical temperature sensor.

**Figure 6 sensors-21-01620-f006:**
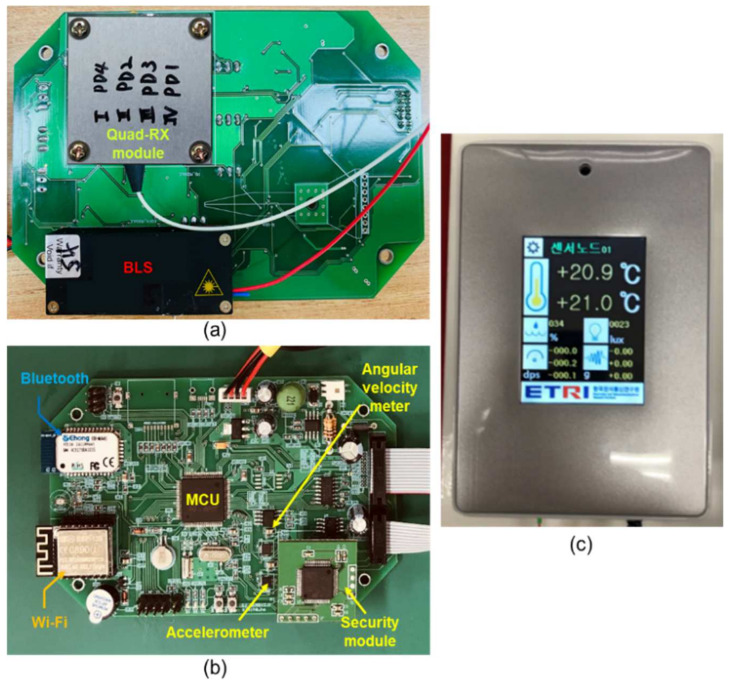
Photographs of the (**a**) top-view of the first layer, (**b**) top-view of the second layer, and (**c**) the end product.

**Figure 7 sensors-21-01620-f007:**
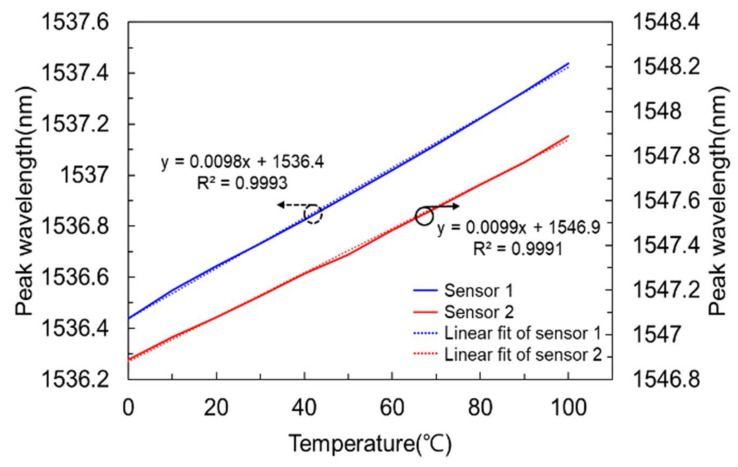
Output characteristics of packaged optical sensors 1 and 2 measured using an optical spectrum analyzer (AQ6319, YOKOGAWA).

**Figure 8 sensors-21-01620-f008:**
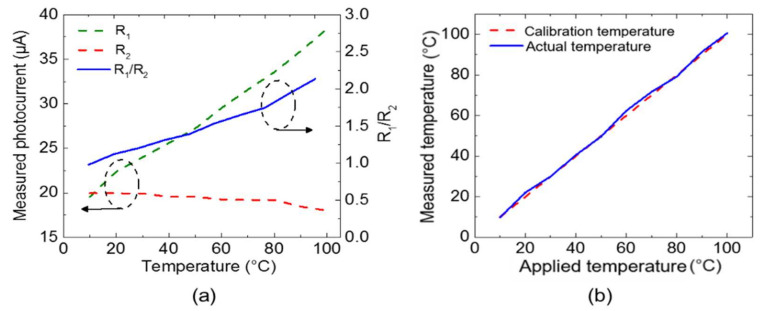
(**a**) Measured photocurrents and R_1_/R_2_ as a function of temperature and (**b**) R_1_/R_2_ dependent temperature data and calibration data.

**Figure 9 sensors-21-01620-f009:**
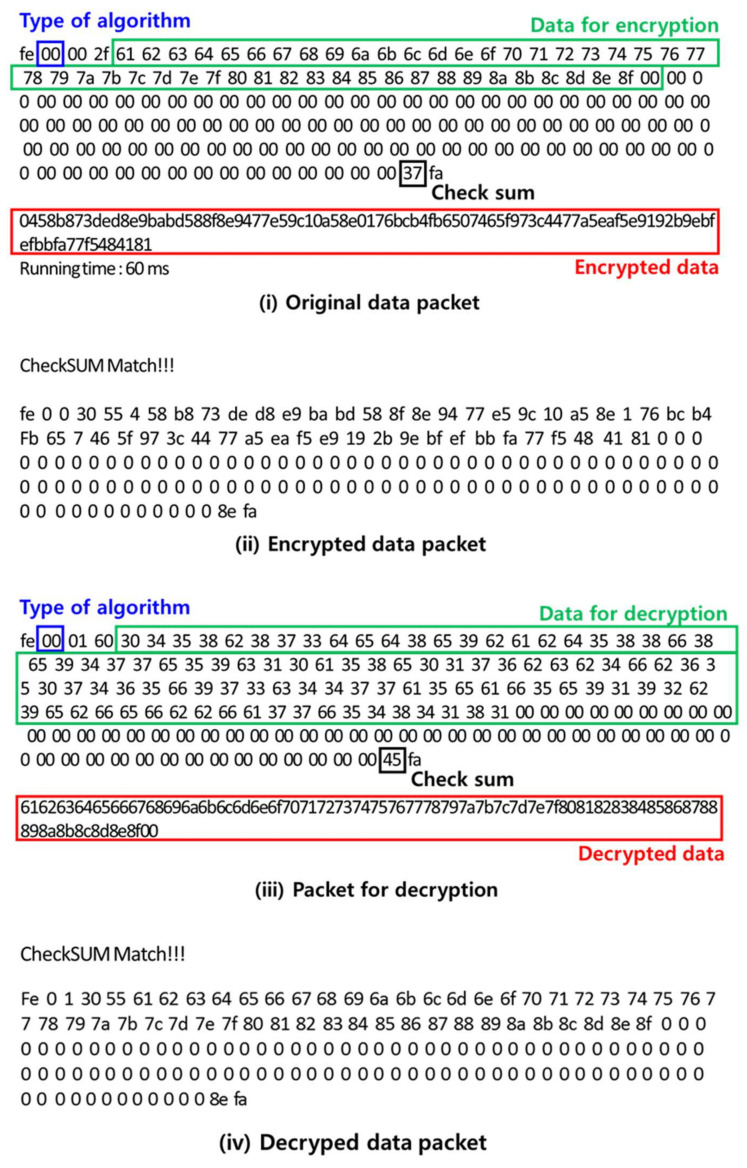
The processes for encryption and decryption between the sensing device and gateway.

**Figure 10 sensors-21-01620-f010:**
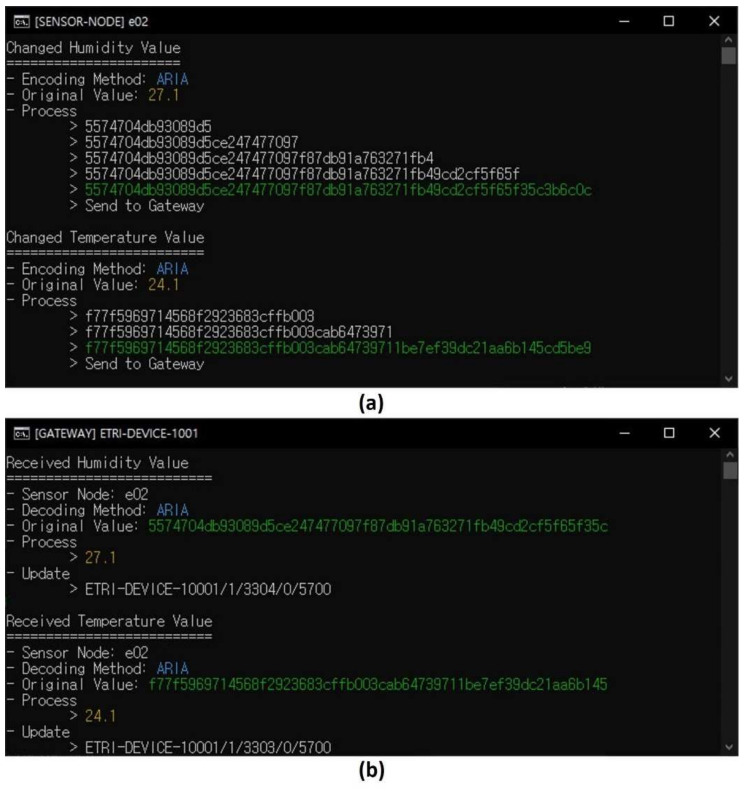
(**a**) Encrypting and transferring data from a sensing device; (**b**) decrypting on a gateway.

**Figure 11 sensors-21-01620-f011:**
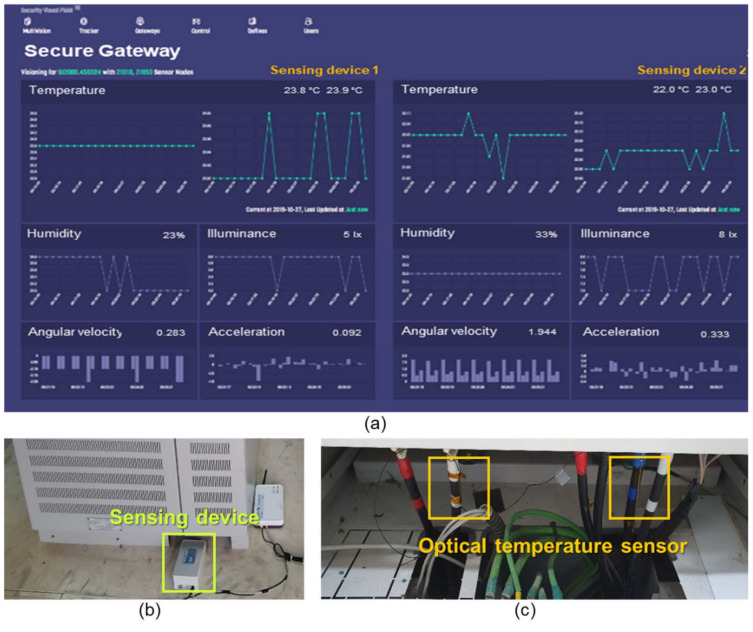
(**a**) Real-time facility monitoring using the developed control platform at the facilities. (**b**) Installed sensing devices and gateway. (**c**) Optical temperature sensors.

**Table 1 sensors-21-01620-t001:** Possible encryption types and key information for the security module.

Algorithm	Operation Mode	Input Length (bits)	Key Information (bits)	Variable Contents
ARIA	ECB	128	RK: 128/MK: 256	RK
HIGHT	ECB	64	RK: 128/MK: 64	MK
	CBC	128	MK: 128/IV: 64	MK, IV
	CTR	No-limit	MK: 128/Counter: 64	MK, Counter
SEED	ECB	128	RK: 256/MK: 128	MK
	CCM	No-limit	MK: 128	MK, etc.
	GCM	No-limit	MK: 128	MK, etc.
SHA	-		-	-

Round key (RK); master key (MK); initial vector (IV); electronic code book (ECB); cipher block chaining (CBC); counter (CTR); counter with CBC-message authentication code (CCM); Galois/counter mode (GCM); academy, research, institute, agency (ARIA); high security and light weight (HIGHT); secure hash algorithm.

## Data Availability

Not applicable.

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
