# Peer review of "A Sensing Device with the Optical Temperature Sensors-Based Quad-RX Module and a Security Module"

_sensors, 2021, doi:10.3390/s21051620_

Round 1
Reviewer 1 Report
The paper presented an end to end solution of optical temperature sensing using FBGs. The presented system includes the sensor, interrogation system and computing and communication electronics. While the completeness and engineering accomplishment of the system presented is high, the level of novelty as a research output is lower. However, it is in my opinion that the work is still interesting to the fiber sensing community. The following are my comments to the authors.
- Please revise the language of the paper. While the paper is well structured, language deficiencies at some points of the paper makes it difficult for readers to comprehend the content.
- It was mentioned that the FBGs were protected using a brass tube and there are 2 FBGs in series for distributed sensing, hence a Quad-RX design. Does this mean that for every FBG added, 2 more output will be required? More discussions on the expansibility of the system will be helpful.
- From the experiment, it is understood that the BF and LF pair allows a ratio to be taken from the outputs that is configured to the temperature change, so light fluctuations from other parts of the optical system do not affect accuracy. However, it was not expressed clearly in the text.
- I would advise that the authors review more of optical interrogation of fiber sensors in general, especially using fiber devices to interrogate such sensors. Some previous works includes using FBGs as band pass filters and slope filters in configuration similar to the work presented to interrogate fiber sensors. A few works are presented in publications and should be reviewed and compared.
Author Response
I appreciate for reviewing this paper.
I am attaching the word file for your reply, so please check it.

Reviewer 2 Report
I am ready to recommend the manuscript for publication. The manuscript is well organized, has a clear presentation, and is scientifically based. However, I would like to add the following remark. The authors did not specify the requirements for temperature stabilization of optical linear filters (LF) and band-pass filters (BF). I would recommend this moment to be reflected in the manuscript because the temperature stabilization of optical filters affects characteristics.
Author Response
I appreciate for reviewing our paper.
I am attaching the word file for your reply, so please check it.

Reviewer 3 Report
The authors present their work on a optical temperature sensor used for monitoring temperature-critical objects and connect to the hub wirelesly.
Reading the paper, several questions popped up and remained unanswered:
- "wavelength change"
It is mentioned several times but is not clear why is it important, and how does it work. Perhaps you mean the shift in the peak of emitted wavelength by the measured object? In any case, this requires a much better description. - "Hardware-based security modules have various advantages over software-based security modules, such as secure storage and management of encryption keys, and rapid encryption/ decryption processing."
Well yes, but do you need those at all? How do you justify using it? Temperature data is likely not sampled with high enough frequency (actually the information about the sampling frequency would be welcome) that speed of encoding would be an issue. Secure storage of keys might not be a big issue either unless firmware is on external flash chip. It might be more convenient to use, but even that is debatable. More details about the used MCU, the required security, etc are required in the paper. - "The BLS provides several benefits, including high stability and the capability of increasing in the number of channels"
What channels do you speak of? - "This leads to low linearity when sensing temperature."
What leads to low linearity? Pulling the sensor? If so, why? What is 'low linearity when sensing temperature'? - "Based on this test, we can confirm that encryption and decryption using the developed security module works reasonably well."
No test was described, only half of the procedure for data transmission. - Fig 8 is not clear at all. Instead of putting in some screenshots, the data could be put in text form.
Moreover, which encryption procedure is shown (it was mentioned earlier that 4 are supported by the HW)?
The encryption/decryption procedure is also not clear to me, as it is mentioned that symmetric-key procedure is used - what information is then stored in the 'decryption packet'?
Is there a problem if the attacker receives both packets?
How are these two packets transmitted, one after the other, via different means, something else? - "we installed sensing devices, gateway and optical temperature sensors"
Isn't the temperature sensor a part of the sensing device? It is difficult for me to decipher whether two temperature senors were used in the described device or only one. There is the Quad-RX, but you mention an optical sensor several times and it s not clear whether this is only the part connected to the Quad-RX via an optical fiber or a separate sensor. - "the optical sensor that is driven without applying power": That's called a passive sensor.
- Why are there accelerometers on the sensing device?
- "a packaged optical temperature sensor and a Quad-Rx module were employed in the sensing device to collect a large number of precise temperature data at various points simultaneously using a single FBG sensor"
What does 'large number of points simultaneously' mean? Points in space - surely not? Points in frequency spectrum - I only see 2?
English must also be greatly improved for the article o become readable. In some places, the used language really makes it hard to follow.
Author Response

(The authors gave the same response as above.)

Round 2
Reviewer 1 Report
Please continue to improve language. There are obvious language errors including misspelling of headings. Otherwise I am satisfied with your response to my comments.
Author Response
I appreciate for your critical review. The paper was revised according to the reviewer's opinion and English correction was performed. The certification was attached to the word file.

Reviewer 3 Report
I have only reviewed the author's response in depth and skimmed the revised manuscript.
In response to 1)
The authors provide a change in a sentence, which now reads:
"The FBG sensor can detect temperature changes by shifting the peak wavelength to 0.01 nm linearly when changing 1°Ð¡."
This sentence is wrong, since the sensor does not shift the peak temperature. My guess guess it estimates the shift in peak wavelength emitted by the measured object, but I am not an expert in this area, os the authors should really carefully fix the description here.
While I believe I can contribute the error to language, such errors are not permissible in a scientific article. Although I did specify that language has to be greatly improved, this has not happened.
In response to 2)
The authors have improved the part significantly.
In response to 3)
The authors did not change much. I am afraid that just adding the word "sensing" does not improve my understanding of the sentence.
In response to 4)
The authors did not change manuscript at all. My questions were not meant for me, they were meant to demonstrate the aspects that the reader could have trouble understanding. Therefore, I expected that the manuscript be improved with the answers to my questions. "Low linearity" in absolute terms means nothing to me. After reading the author's answer I believe the authors meant that the linearity of the response will be worse it sensor is put under stress. Therefore the authors should just write that. Perhaps they could write that the sensor should be fixed in a way that produces as little stress as possible in (I am guessing here) axial direction, since any stress would decrease the accuracy of the measurements. If that is indeed what they meant.
In response to 5)
The authors modified the figure but that was not what I expected. The term "test" must be either wrong here as the authors describe no test prior to the pointed out sentence. I believe now they performed an experiment in which they prescribed a packet, encrypted it, then decrypted it, and confirmed the resulting data matched the original packet. And that's what the author's base their claim of the encryption working "reasonably well" on. That is just a demonstration of how the encryption works, it is not nearly a test of how well it works.
In response to 6)
The modified figure and accompanying text is a bit clearer now. The reader is stil required to perform far too much reasoning, regarding what is written. It seems the authors are describing the packets of data transferred between the MCU and the encryption module, while I was initially under the impression they describe the data that are transmitted over the wifi. This is still only my speculation though, and the authors should improve the text further.
In response to 7)
The manuscript is not much improved. While I now understand the measurement methodology - it requires a FBG-based sensor to be attached to the object whose temperature is being measured, and it requires an illumination source (the sensor measures reflection, not radiance), I came to understand it through external sources, not through the presented manuscript. Other details I am still unclear un - Does the sensing device contain multiple temperature sensors (i.e. can it sense multiple temperatures simultaneously), what do "channels" mean in the manuscript (is a channel a temperature measurement, a wavelength measurement - are there multiple wavelengths measured per single temperature measurement, or something else)
In response to 8)
OK
In response to 9)
Again the authors reply only to me, and do not modify the manuscript. If the authors explicitly point at the accelerometer (and also Bluetooth for that matter), on a figure, then they should mention them in the manuscript even if only to tell the reader that these are for future use. Although it is now clear from added figures that the accelerometer is already used but is just ignored in the manuscript. There is also illuminance and humidity - are these sensors on the device as well? Why not put in a paragraph that describes the device in short or just exclude everything that is not relevant to the present manuscript?
In response to 10)
The authors do not explain what does 'large number of points simultaneously' mean. Again they add the term "multi-channel", which does not help much. What does multi-channel mean in this context? The sentence implies there are multiple temperature readings taken by the sensor. This, I still believe is false but cannot confirm.
In response to 11)
The advancements in language are not sufficient. The manuscript requires quite an effort from the reader to read. A bigger problem, however, is that the poor language causes some sentences to get a very different meaning.
Author Response
I appreciate for your critical review.
The paper was revised according to the reviewer's opinion and English correction was performed.
Please refer to the attached file.
